# The Applicability of a 2-Transcript Signature to Identify Bacterial Infections in Children with Febrile Neutropenia

**DOI:** 10.3390/children10060966

**Published:** 2023-05-29

**Authors:** Johannes Aasa, Eva Tiselius, Indranil Sinha, Gunnar Edman, Martina Wahlund, Shanie Saghafian Hedengren, Anna Nilsson, Anna Berggren

**Affiliations:** 1Division of Pediatric Oncology, Department of Women and Children’s Health, Karolinska Institutet, 17177 Stockholm, Sweden; johannes.aasa@stud.ki.se (J.A.); eva.tiselius.2@ki.se (E.T.); shanie.hedengren@ki.se (S.S.H.); anna.berggren@ki.se (A.B.); 2Department of Clinical Sciences, Karolinska Institutet, 17177 Stockholm, Sweden; gunnar.edman@gmail.com; 3Research and Development, Norrtälje Hospital, 76145 Norrtälje, Sweden; 4Sanofi Sverige, Franzeng 6, 11251 Stockholm, Sweden; martina.wahlund@gmail.com; 5Division of Pediatric Hematology-Oncology, Tema Barn, Astrid Lindgren Children’s Hospital, 17164 Solna, Sweden

**Keywords:** children, febrile neutropenia, bacterial infections, 2-transcript signature, *FAM89A*, *IFI44L*

## Abstract

Febrile neutropenia is a common complication during chemotherapy in paediatric cancer care. In this setting, clinical features and current diagnostic tests do not reliably distinguish between bacterial and viral infections. Children with cancer (n = 63) presenting with fever and neutropenia were recruited for extensive microbiological and blood RNA sampling. RNA sequencing was successful in 43 cases of febrile neutropenia. These were classified as having probable bacterial infection (n = 17), probable viral infection (n = 13) and fever of unknown origin (n = 13) based on microbiological defined infections and CRP cut-off levels. RNA expression data with focus on the 2-transcript signature (*FAM89A* and *IFI44L*), earlier shown to identify bacterial infections with high specificity and sensitivity, was implemented as a disease risk score. The median disease risk score was higher in the probable bacterial infection group, −0.695 (max 2.795; min −5.478) compared to the probable viral infection group −3.327 (max 0.218; min −7.861), which in ROC analysis corresponded to a sensitivity of 0.88 and specificity of 0.54 with an AUC of 0.80. To further characterise the immune signature, analysis of significantly expressed genes and pathways was performed and upregulation of genes associated to antibacterial responses was present in the group classified as probable bacterial infection. Our results suggest that the 2-transcript signature may have a potential use as a diagnostic tool to identify bacterial infections in immunosuppressed children with febrile neutropenia.

## 1. Introduction

For paediatric patients presenting with symptoms of infectious disease, determining the aetiology has been a long-standing challenge for physicians. The emergence of antibiotic resistance has brought to light a more acute need for rapid and reliable diagnostic methods [1,2]. Blood culture analysis, although considered the gold standard for diagnosing bacterial infections, is time-consuming and offers a level of sensitivity which often is unsatisfactory. This is especially true among paediatric patients for whom low blood volumes contribute to fewer positive findings compared to the adult population. With the introduction of polymerase chain reaction (PCR) techniques, the possibility of rapid and high-sensitivity screening for viral, and in some cases bacterial, pathogens have emerged [1,3].

In high-risk populations, such as immunodeficient children with febrile neutropenia (FN), the limitations of today’s diagnostic methods are even more evident. FN is a common complication upon treatment with myelosuppressive drugs and the main cause of treatment-related morbidity in paediatric cancer care [4,5,6]. Therefore, the universal recommendation for managing FN is empirical antibiotic treatment for all patients [7,8]. Previous studies investigating the aetiology of FN in children have suggested that viral infections could be a more common finding in FN than bacterial infections [9,10,11]. Several biomarkers such as C-reactive protein (CRP), procalcitonin and IL-6 have been introduced into clinical decision making to discriminate between cases caused by bacterial or viral infection [12,13]. However, the diagnostic accuracy of these biomarkers varies between studies where, for example, CRP has a high sensitivity but low specificity for bacterial infections [13,14]. Thus, none of the above-mentioned markers alone can accurately differentiate bacterial from non-bacterial infections in FN patients.

In recent years, host transcriptome analysis has delivered promising results with regards to identifying bacterial infections among febrile patients [15,16,17]. The study by Herberg et al. [18] showed that bacterial infections can be identified with 100% sensitivity and 96% specificity in febrile children, using a 2-transcript signature comprised of the genes family with sequence similarity 89, member A (*FAM89A*) and interferon-induced protein 44-like (*IFI44L*). At present, it is not known whether host transcriptome analysis is applicable as a diagnostic tool in immunosuppressed children. We previously showed that un-biased host transcriptome analysis in children with FN did not clearly associate microbiologically defined infections (MDI) to expected innate immune-associated pathways, but rather to cellular stress responses and cell proliferation [19]. More recently, it was shown that FN children with bacteraemia presented with specific blood transcriptome immune signatures compared to FN children with other MDIs [20]. In adult FN patients, classifiers consisting of metabolomic and gene expression data were shown to accurately predict bacteraemia when analysing genes related to innate immune responses [21]. The combined data, albeit from a small number of studies, point to high throughput “omics” as a potential tool for the discovery of biomarkers in FN patients.

In this study, we investigated if the 2-transcript signature could be used also to discriminate bacterial infections in the same cohort of FN children as previously published [19]. Firstly, samples were analysed using RNA sequencing to determine if patterns of *FAM89A* and *IFI44L* activation together with clinical data could discriminate between viral and bacterial infection in FN children. Secondly, RNA-sequencing data were analysed regarding all significantly expressed genes to compare if these followed the patterns expected for viral and bacterial infections.

## 2. Materials and Methods

### 2.1. Subjects

The Regional Ethics Board in Stockholm approved the study with number: 2008/648-31/4 and 2009/286-32. A prospective, observational study on children aged 0–18, who were enrolled at the Childhood Cancer unit at Astrid Lindgren Children’s Hospital between January 2013 and June 2014 was conducted. When presenting with symptoms of FN, patients and their parents/caretakers were approached for study participation and gave informed consent.

### 2.2. Case Definition and Sampling Procedures

FN was defined as follows: a body temperature ≥ 38.5 °C on one occasion or ≥38.0 °C on two occasions at least 60 min apart, combined with an absolute neutrophil count of either ≤0.5 × 10^9^/L on one occasion or ≤1.0 × 10^9^/L with a decline to less than ≤0.5 × 10^9^/L over a subsequent 48 h period. Children were included and sampled within 72 h of fever onset, as previously described in detail [19]. Other clinically relevant data were accessed via the electronical patient chart. A child with recurring episodes of FN could be enrolled again provided that the child had been afebrile for >72 h and was no longer receiving antibiotics for the previous FN episode. After recovery from the FN episode, the child was invited to leave a further set of test samples at their next scheduled visit to the hospital to serve as control samples.

### 2.3. Sample Collection and Storage

Sampling was performed for all cases at inclusion in the study. Blood RNA samples were collected in PAXgene Blood RNA Tubes (PreAnalytiX, Homebrechtikon, Switzerland) and stored at −20 °C. Nasopharyngeal aspirates (NPA), blood cultures and clinical chemistry were collected for all FN episodes. Additional microbiological sampling was performed when clinically indicated. Analysis of the NPA samples and blood cultures were analysed at the Karolinska University Laboratories as previously described [19].

### 2.4. RNA Sequencing and Analysis

The purity and concentration of the extracted RNA was measured using RNA ScreenTape Assay (on an Agilent 2200 Tapestation, Agilent Technologies, Santa Clara, CA, USA) and NanoDrop ND-1000 spectrophotometer (ThermoFisher Scientific, Waltham, MA, USA). An RNA integrity number (RIN) > 7 was considered a good sample quality.

For sequencing library construction, an Illumina TruSeq Stranded kit (San Diego, CA, USA) was used. In brief, mRNA isolation, RNA fragmentation, cDNA synthesis, adapter ligation and cDNA amplification were performed according to the manufacturer’s protocol. The resulting cDNA library was normalized, pooled and sequenced. Sequencing was done using Illumina Nextseq 550 with 75-cycle v2 sequencing kit which generated 75-bp single-end reads. The reads were then mapped to the human reference genome, GRch38, and normalized using the DESeq2 normalized method. SeqMonk (version 1.47.1) was used to identify differentially expressed genes between groups and controls, with a false discovery rate (FDR) < 0.05 and a fold change (FC) difference of at least 2 as the criteria for significance. Pathway analysis was performed using QUIAGEN IPA (QIAGEN Inc., https://digitalinsights.qiagen.com/IPA, URL accessed on 24 April 2023).

Raw data was assessed at the GEO database, accession number; GSE152341 as previously described [19].

### 2.5. Definition of Test Groups

#### 2.5.1. Test Groups Based on Microbiologically Defined Infections

After initial microbial testing, participants were grouped according to the MDI into four categories: bacterial infection (BI), viral infection (VI), co-infections and unknown etiology as previously described [19]. Patients with a clinically relevant bacterial finding in blood or localized infections with a positive bacterial culture (urine, skin) were categorised as BI. Patients with a positive virus-specific PCR finding were classified as VI. Patients with a bacterial finding as above and a positive virus-specific PCR were classified as “co-infections”. Participants for whom no MDI was detected during the FN episode were classified as unknown etiology.

#### 2.5.2. Test Groups Based on Microbial Detection and CRP Cut-Off Values

The microbial categorization was the same as described above, but with the addition of a set of CRP cut-off values. The test groups BI and co-infections were grouped together with cases with unknown etiology and VI group who had a CRP > 100 mg/L at any point during the FN episode to form a new group termed probable bacterial infections (PBI). All cases with unknown etiology with CRP < 20 mg/L or below were grouped together with the remaining VI, forming a new group termed probable viral infection (PVI). The remaining cases with unknown etiology were grouped as FUO (fever of unknown origin). In this way, three groups were formed and subjected to analysis PVI, PBI and FUO.

#### 2.5.3. Statistical Analysis

For descriptive statistical analysis, Kruskal–Wallis test and Dunn’s multiple comparisons test were carried out for continuous data whereas a Chi-square test was used for categorical data. A *p*-value ≤ 0.050 was determined statistically significant. Blood counts registered as <0.1 were counted as 0.1 in statistical analyses; time points registered as >30 days were included as 30 throughout analysis. Calculations were performed in GraphPad Prism (GraphPad Prism, San Diego, CA, USA). To limit the occurrence of type 1 errors originating from the sequencing procedure, the Benjamini–Hochberg method was used. False discovery rate (FDR) was set to <0.05.

To assess the difference in gene expression between *FAM89A* and *IFI44L* a disease risk score (DRS) was used as previously described [18,22]. DRS was calculated by subtracting the normalized log2 expression value of *IFI44L* from the normalized log2 expression value of *FAM89A*. A higher DRS was hypothesized as indicative of bacterial infection and a lower DRS as indicative of viral infection. To test if the DRS obtained could discriminate between infections, Receiver Operating Characteristic (ROC) curves and Area under the curve (AUC) were calculated. Finally, the optimal DRS cut-off was analysed using the ROC curves and applied to all study cases to assess how many of the total number of cases the 2-transcript signature would identify as bacterial infections. The optimal DRS cut-off was selected from the ROC curve where DRS offered optimal specificity and sensitivity.

## 3. Results

### 3.1. Cohort Characteristics

Of the 67 episodes that met the study criteria, four were excluded due to RNA samples being collected later than the allocated 72 h after fever onset. A further 20 episodes were then excluded from the study because the blood samples contained insufficient amounts of RNA. In total, 43 episodes consisting of 35 children were included in the study groups (Figure 1). Twelve children were sampled after their FN episode and served as controls.

Initially, FN episodes were classified according to MDI after which two different CRP cut-offs were used for reclassification to either PBI, PVI or FUO (Figure 1). Fourteen episodes in total were re-allocated to a different test group with three episodes added to the PVI group because of a CRP < 20 g/L and 11 episodes with a CRP > 100 g/L were re-allocated to the PBI group. A detailed description of the clinical characteristics of these episodes is shown in Appendix A. Incorporating CRP in the categorization created three groups of a similar size with PBI (n = 17), PVI (n = 13) and FUO (n = 13). Patient characteristics of the respective groups are shown in Table 1. A significant difference (*p* ≤ 0.05) was observed in peak CRP as expected, when comparing PBI to PVI as well as PBI to FUO. In addition, days hospitalized and days with antibiotics differed significantly when comparing PVI and PBI. Respiratory symptoms were most common in PVI compared to PBI and FUO groups.

The expression data of the two genes of interest, *FAM89A* and *IFI44L*, were then combined into a disease risk score (DRS). In the first analysis based on MDI, the highest DRS were found in the bacterial group with a median of −1.253 (max 0.883; min −3.398) followed by the FUO group. All four test groups displayed DRS values with an overlapping interquartile range to all other groups. ROC analysis revealed an AUC of 0.71. Optimal values for sensitivity and specificity were 1.0 and 0.47, respectively. These were obtained when the cut-off value for DRS was set at −3.5.

To further explore the ability of DRS to discriminate between PBI, PVI or FUO in FN cases, the DRS score was applied to the new test groups where the CRP cut-offs were part of the classification. The median DRS was higher in the PBI group, −0.883 (max 2.795; min −7.835) compared to the PVI group −3.646 (max 0.218; min −7.861) and the FUO group −3.776 (max 2.911; min −5.503. There was a significant difference for the DRS score comparing PBI to PVI (*p* = 0.0178). (Figure 2A). The performance of the DRS in this grouping showed that a cut-off for DRS of −3.5 optimally separated the two groups. In the ROC analysis, this corresponded to a sensitivity of 0.88 and specificity of 0.54 with an AUC of 0.80 (Figure 2B). We then applied the DRS score to our total cohort of 43 episodes of FN and interestingly, 27 episodes (63%) were then classified as having a PBI compared to the initial 17 episodes (49%). Notably, 100% of the FN episodes received broad-spectrum antibiotics.

### 3.2. Analysis of Other Significantly Expressed Genes and Pathways

To further characterise the immune signature of FN cases, a pathway analysis was performed. A list of the top 20 significantly expressed genes (up- and downregulated) can be found in Appendix A. There were 27 significantly upregulated genes and 10 significantly downregulated genes when matching PBI against controls. The three genes *MPO*, *ELANE* and *CTSG* commonly associated with antibacterial immunity were upregulated. Pathway analysis attributed the activation of these genes to two of the top five canonical pathways, which are both associated with neutrophil antibacterial activity (Figure 3). Among the downregulated pathways in this analysis, several canonical pathways related to interferon signalling and antiviral activity were found. When matching PVI against controls, 1202 genes were significantly expressed, of which 241 were upregulated and 961 genes were significantly downregulated. A pathway analysis revealed that out of the top-ranked upregulated canonical pathways, all were related to cell proliferation while the downregulated canonical pathways were associated with inflammatory responses. Hence, no pathways commonly associated with an antiviral response or any other antimicrobial response were upregulated, nor were any of the top five downregulated pathways related to any antiviral or antibacterial response (Figure 3).

Next, matching the PBI against the PVI group, 467 genes were significantly upregulated and 27 genes significantly downregulated. Among the upregulated genes, *PRTN3* was expressed with the highest fold change (log2 FC 5.298) and several other genes associated with an antibacterial response were highly upregulated (log2 FC > 3.0). Among the downregulated genes, significantly expressed genes such as *OAS3* and *SAMD4A* were identified as having a clear role in the antiviral response. Pathway analysis revealed four of the top five upregulated canonical pathways bearing some relevance in the antibacterial response, and among the top five downregulated canonical pathways, three pathways were associated with an antiviral response (Figure 3). Finally, the FUO was compared to PVI, PBI and controls. There were only six genes upregulated compared to the PBI group and no further analysis was done. When FUO was compared to controls, genes implicated in cell proliferation and inflammatory response were seen in the upregulated and downregulated pathways. A similar finding was noted when comparing PVI with controls. When comparing FOU with PVI, two upregulated pathways overlapped with the PBI vs. PVI analysis and when focusing on the top five downregulated pathways, 17 genes correlating to the T-cell receptor clustered into these pathways. Similar to PBI, the genes MPO and CTSG were found to be upregulated when matched against controls (Figure 3).

## 4. Discussion

In this study, we investigated whether whole-blood RNA expression of the genes *FAM89A* and *IFI44L* from children with FN could be used as a diagnostic tool to discriminate bacterial from viral infections. In addition, significantly up- and downregulated genes and pathways were analysed to understand if these immune signatures supported the 2-transcript signature in separating bacterial and viral infections. The 2-transcript signature as a diagnostic tool performed with an AUC of 0.80 when test groups were defined using microbiologically defined infections and CRP in combination.

Two different criteria for categorizing were used. One was based on MDI only and the other on MDI combined with CRP cut-offs. Studies have shown that using a CRP cut-off > 80 mg/L to rule in a bacterial infection and a CRP cut-off < 20 mg/L to rule out bacterial infections is reliable with >80% sensitivity and >90% specificity [23,24]. It has also been shown that CRP as a predictor of bacterial infection is comparable in FN cases and in cases without immunosuppression, albeit with a lower specificity in FN [13,25]; which is why we choose to apply a stricter CRP cut-off with an upper limit > 100 mg/L to rule in bacterial infections. With these CRP limits, the DRS score for PBI was significantly higher than for PVI, but not FUO. The DRS reached significant AUC values for the test group based both on CRP and MDI. The lower specificity of the 2-transcript signature encountered in the test group only using MDI may be a consequence of microbiological misclassification using conventional diagnostics such as blood culture that are known to perform with low sensitivity [1]. It is also possible that some of the viral findings in FN cases were remnants of previous asymptomatic infections as previously shown to occur in children [26].

Our data on the 2-transcript signature suggests that this signature may be used as a diagnostic tool in children with FN as well. However, both the sensitivity and specificity were lower than in the original study in immunocompetent children [18] and the reasons for this discrepancy can be numerous. The performance of the signature is dependent on correct classifications of microbiological infections which could be more challenging in immunosuppressed children who have an increased risk for polymicrobial infections [27] as well as infections originating from commensal flora from the skin or gut [28]. In addition, children with FN often present with minimal clinical signs of severe infection due to the blunted immune response [29]. In the present study, RNA was extracted from white blood cells that were exposed to cytotoxic drugs which may affect the expression of relevant genes.

Studies on blood transcriptome profiles in children presenting with FN are few. In a study published by us [19] using an unbiased approach, we were not able to detect a pathogen-specific immune signature in children with FN similar to that already published for otherwise healthy children with MDI [18]. In a more recent study on blood transcriptome profiles in children with FN, approximately 500 genes were identified that differed between episodes with a documented MDI as compared to episodes without documented infection [20], and pathway analyses revealed differences in genes involved in phagocytosis and lysosome/vesicle formation as part of the innate immune response.

In the current study, several genes involved in bacterial defence were among the most upregulated and clustered into pathways involved in neutrophil antibacterial activity. More specifically, neutrophil serine proteases such as the *ELANE*, *CTSG* and *PRTN3*, all among the most upregulated genes when comparing PBI to both controls and PVI, contribute to antimicrobial defence by attacking membrane-associated or capsule proteins in bacteria [30,31]. It is noteworthy that these neutrophil-associated immune responses appear to dominate the antibacterial response, even in a patient population where neutrophil levels are abnormally low. Thus, even in these severely immune-suppressed children, a clear antibacterial response was detected in the PBI test group which supports the accuracy of the 2-transcript signature. The FUO test group only differed by six genes to the PBI group. In addition, when focusing on the downregulated genes, several genes involved in interferon signalling and mediating viral defence were downregulated in the PBI test group. We expected to find an upregulation of these same genes in the PVI test group, but this was not the case. Instead, when comparing PVI against controls, no viral specific signature was noticed, and pathway analyses of the upregulated genes indicated that these were involved in cell cycle regulation. When comparing FUO to the PVI test group, genes related to T cell signalling and effector functions were noted.

There are several limitations to this study that we must acknowledge. First, the small cohort size suggests that all conclusions arrived at should be evaluated in larger future studies. Secondly, the classification of BI was a mix of cases with invasive bacteraemia and cases with localized infections. In a more homogenous cohort with severe invasive infections, the immune signature may be more distinct even in an immunosuppressed host. The number of cases excluded from analysis due to insufficient amounts of RNA available was high and exceeded that reported from other studies [20,21].

In short, the results presented herein suggest that the 2-transcript signature proposed by Herberg et al. may have a potential use as a diagnostic tool to identify bacterial infections in children with febrile neutropenia. However, our findings must be confirmed in larger studies to assess the clinical utility of such a diagnostic tool.

## Figures and Tables

**Figure 1 children-10-00966-f001:**
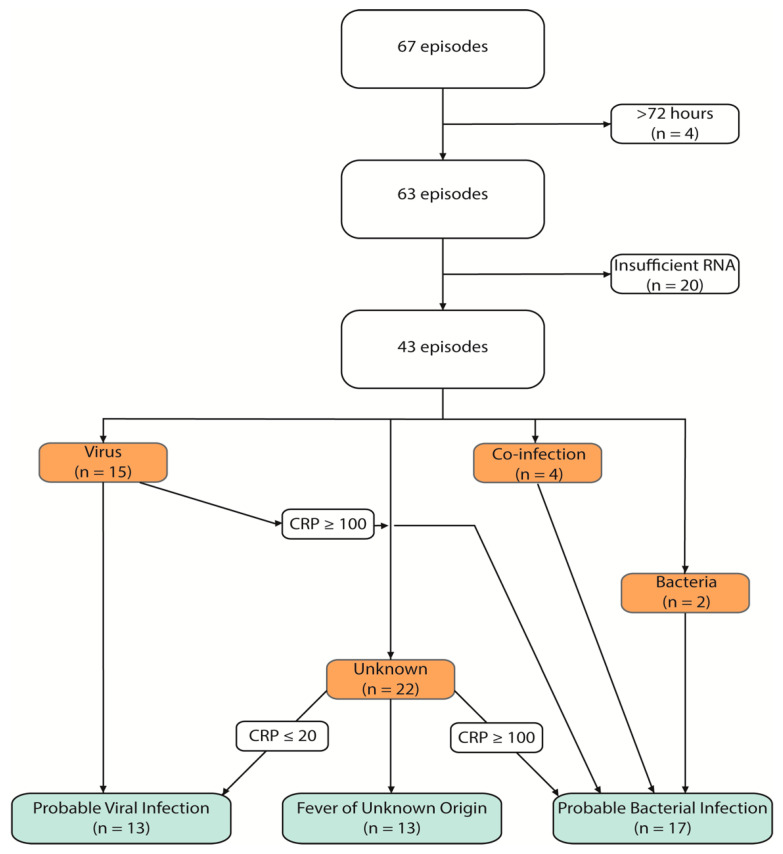
Flow chart depicting classification of cases based on microbiological finding and c-reactive protein (CRP) levels. Sixty-seven episodes of febrile neutropenia were initially included. Four episodes were excluded due to sampling >72 h from onset of fever; 20 were excluded due to insufficient RNA levels. The remaining 43 episodes were initially categorized in [19] based on microbiological findings, colored in orange. In this study, we categorized the episodes further, based on the levels of C-reactive protein (CRP), seen in green.

**Figure 2 children-10-00966-f002:**
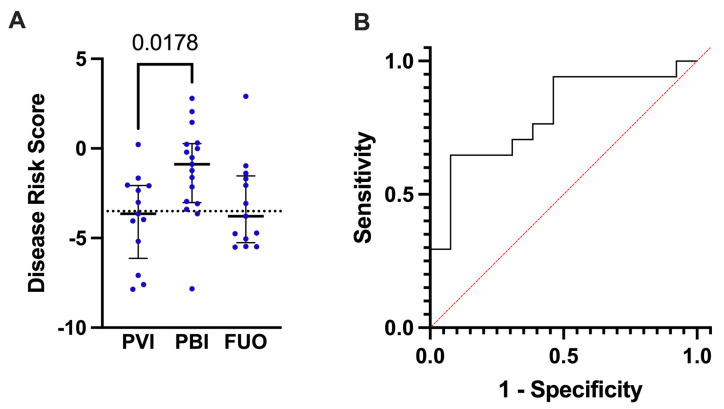
(**A**). Distribution of disease risk scores (DRS) in the probable viral infection (PVI) group, the probable bacterial infection (PBI) group and fever of unknown origin group. Midlines represent the median; error bars extend over the interquartile range. Individual values are plotted as blue dots. Dotted line represents the cut off-value at −3.5, which separates between predicted bacterial and predicted viral infections. (**B**) Receiver operating characteristic (ROC) curve illustrating the ability of the disease risk score (DRS) to discriminate between PBI and PVI in children with FN. The area under curve (AUC) equals 0.80.

**Figure 3 children-10-00966-f003:**
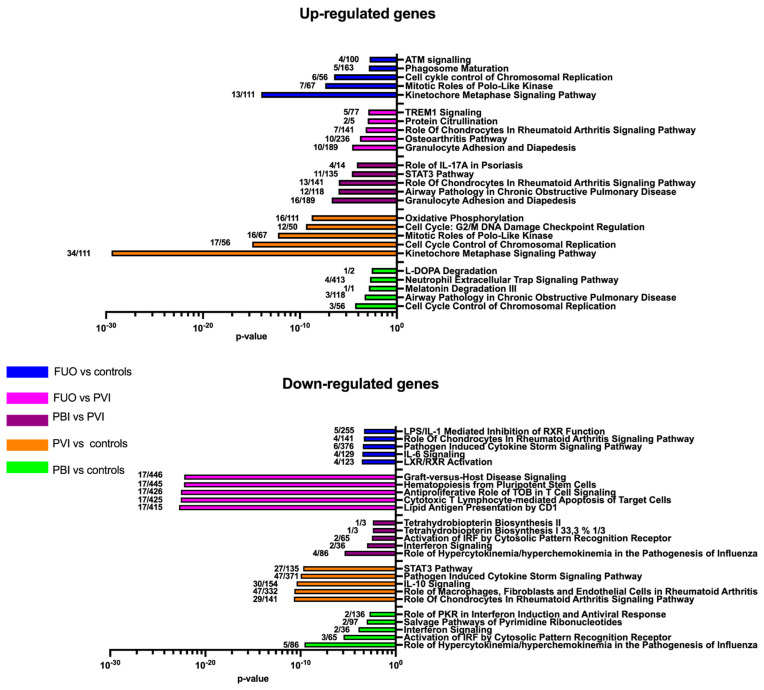
Pathway analysis for PBI, PVI and FUO test groups. The columns represent the top five up- and downregulated significant canonical pathways for each group after Ingenuity Pathway Analysis. The numbers after the columns indicate number of genes up- and downregulated in each network.

**Table 1 children-10-00966-t001:** Clinical characteristics of the FN episodes in the different groups. Statistical analysis on continuous and categorical data was carried out on the probable viral infection (PVI), probable bacterial infection (PBI) and fever of unknown origin (FUO) groups. Episodes where patients had ANC < 0.1 were counted as 0.1 in statistical calculations; timepoints registered as >30 were counted as 30.

	Probable Viral (n = 13)	Probable Bacterial (n = 17)	Fever of Unknown Origin (n = 13)	Control (n = 12)
**Age (years)** **(median, range)**	4.6 (0.6–15.6)	9.4 (0.5–16.1)	9.7 (0.5–15.7)	9.7 (0.6–15.8)
**Gender (n female) (%)**	8 (62)	10 (59)	5 (38)	8 (67)
**Peak CRP (mg/L) *** **(median, range)**	25 (4–97)	141 (22–412)	59 (28–90)	N/A
**WBC count (10^9^/L)** **(median, range)**	2.1 (0.2–5.6)	1.3 (0.3–14.6)	1.8 (0.2–8.1)	3.7 (0.7–6.6)
**ANC (10^9^/L)** **(median, range)**	0.2 (0.1–0.6)	0.1 (0.1–0.5)	0.1 (0.1–0.4)	2.1 (0.1–5.6)
**Days with neutropenia** **(median, range)**	10 (5–37)	6 (1–>30)	9 (5–20)	N/A
**Peak temperature (°C)** **(median, range)**	38.9 (38.1–40.5)	39.2 (38.1–40.3)	39.3 (38.5–40.2)	N/A
**Days with fever** **(median, range)**	2 (1–6)	2 (1–16)	2 (1–4)	N/A
**Days hospitalized ^†^** **(median, range)**	4 (0–8)	6 (3–>30)	4 (3–10)	N/A
**Days with antibiotics** **(median, range)**	5 (0–10)	8 ^†^ (5–30)	7 (3–10)	N/A
**Respiratory symptoms, n (%) ^‡^**	12 (92)	14 (82)	6 (46)	N/A
**Gastrointestinal symptoms, n (%)**	2 (15)	3 (18)	2 (15)	N/A
**Local symptoms, n (%)**	1 (8)	4 (24)	2 (15)	N/A

Abbreviations: WBC, white blood cell; ANC, absolute neutrophil count. * *p* ≤ 0.05 following Kruskal–Wallis test between PBI and PVI as well as between PBI and FUO. ^†^
*p* ≤ 0.05 following Kruskal–Wallis test between PVI and PBI. ^‡^
*p* ≤ 0.05 following a Chi-square analysis for difference in distribution between groups PVI, PBI and FUO. Disease Risk Score analysis.

## Data Availability

Raw data on RNA expression can be assessed at the GEO database, accession number; GSE152341.

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
