# Peer review of "The Applicability of a 2-Transcript Signature to Identify Bacterial Infections in Children with Febrile Neutropenia"

_children, 2023, doi:10.3390/children10060966_

Round 1
Reviewer 1 Report
While the scientific merit and the findings hopeful it does not seem in keeping with the aim of the journal.
Author Response
We thank the reviewer for reading our manuscript.
However, we fail to understand what the reviewer comment on.
Reviewer 2 Report
The manuscript by Aasa, et al, presents a subsequent analysis of data generated from a previously published RNA sequencing study of pediatric cancer patients. Here, the authors have taken a 2-gene score reported to be associated with bacterial infections in immunocompetent children and applied it to their cohort of immunocompromised children. It has been previously shown that biomarkers like CRP, which are routinely used for clinical decision making, are less accurate in immunocompromised patients. The authors show the same result here for the 2-gene score. Comparisons of differentially expressed genes between different groups was not particularly revealing. The authors focus most on groups classified as either probable bacterial or probable viral and do not do much with the unknown group. One could argue, however, that these are patients in most need of new biomarkers. They should be included in the gene expression analyses.
1. The diagram in Figure 1 and the text Line 209 state “43 children” but should be 35 children or 43 episodes.
2. For Table S1, please indicate patients A-I and J-N in the table since these letters are mentioned in the caption.
3. For Figure 2A, please check the minumum value for DRS for the PBI group stated in the text line 205. The figure shows a value in the same range as the minimum value for the PVI group.
4. Regarding the text in lines 209-211, please provide a figure or table showing the DRS scores for patients in the FUO category. Likewise, please provide a list of genes that are differentially expressed in the FUO v. other groups.
5. In the discussion please address the following: How might the expression of these two genes be affected by neutropenia and/or lymphopenia? Are they expressed by cell types that are abnormally low in FN cases?
Author Response
We thank the reviewer for many constructive comments that we have addressed. Please see below
1.The diagram in Figure 1 and the text Line 209 state “43 children” but should be 35 children or 43 episodes.
We agree with the reviewer. We have changed Fig 1 to episode. We have revised the test througout the manuscript so that episodes and patients are clearly separated.
- For Table S1, please indicate patients A-I and J-N in the table since these letters are mentioned in the caption.
Table S1 has been revised and the nomenclature A-I and J-N has been removed for clarity.
- For Figure 2A, please check the minumum value for DRS for the PBI group stated in the text line 205. The figure shows a value in the same range as the minimum value for the PVI group.
Figure 2A has been altered significantly to also encompass the group FUO as requested. We have checked the text and corrected the values shown in the text.
- Regarding the text in lines 209-211, please provide a figure or table showing the DRS scores for patients in the FUO category. Likewise, please provide a list of genes that are differentially expressed in the FUO v. other groups.
Thank You for this important and constructive comment. We have reanalysed and recalculated the DRS score with 3 groups; PBI, PVI and FUO as requested. This is now shown in Figure 2A:
We have also up-dated Figure 3 with data on the up- and down-regulated genes in the FUO group.In addition, supplemental Table 2 has been updated with a gene list for the FUO group as requested .
- In the discussion please address the following: How might the expression of these two genes be affected by neutropenia and/or lymphopenia? Are they expressed by cell types that are abnormally low in FN cases?
The reviewer raises an important but difficult question to answer. It is known that FAM89A is expressed in basophils, and to a lower extent in neutrophils. IFI44L is reported to be expressed in immune cells not otherwise specified.
In a paper by Hu et al PNAS 2013, the association between gene expression levels and total blood cell count was addressed. The overall conclusion was that total WBS was not associated to the expression of certain genes. However, clusters of genes were associated to neutrophil and lymphocyte counts with border line significance.
Reviewer 3 Report
this study shows the use of a 2-transcript signature to identify bacterial infections in children with febrile neutropenia.
we are surprised with the dates of inclusion of cases in this study between January 2013 and June 2014.
Could the authors precise when these analyses were performed?
The power of discrimination seems not to be sufficient in these immunodepressed patients to be used as a diagnostic tool.
Author Response
We thank the reviewer for reading the manuscript and provided us with comments. Please see response below.
1.We are surprised with the dates of inclusion of cases in this study between January 2013 and June 2014. Could the authors precise when these analyses were performed?
This project was part of a PhD project for a young medical doctor who started her work in 2013 in parallel with clinical rotations. The analysis was performed in 2019 and the first manscript was published in 2020 ( Ref 19 Wahlund et al). the current project and re analysis of the data was performed in 2022 by first author J Aasa in collaboration with bioinforatician I Sinha.
- The power of discrimination seems not to be sufficient in these immunodepressed patients to be used as a diagnostic tool.
Our data were generated in a small cohort of children with FN episodes where bacteremia cases were few. Still, the DRS score discriminates between PBI and PVI. We believe that this score may have a potential use in the clinical setting. However, larger studies are warranted before we can definitely rule in-or rule out this tool why we believe that it is important to publish our findings.
Round 2
Reviewer 2 Report
Thank you for including more data for the FUO group. I have no additional suggestions.